# MMCHAT: A MULTI-TURN MULTI-MODAL CONVERSATIONAL BENCHMARK GROUNDED IN REAL WORLD USER BEHAVIORS

## ABSTRACT

Large Vision-Language Models (LVLMs) are the foundation of modern AI assistants, which users typically engage in multi-turn conversations involving interrelated questions about images. However, existing benchmarks predominantly evaluate LVLMs in isolated, single-turn settings, creating a significant gap with real-world use. While recent efforts have explored multi-turn evaluation, they often rely on oversimplified, synthetic dialogues that fail to capture the complex, context-dependent nature of real-world user interactions. To address this critical gap, we introduce MMCHAT, a multi-turn, multimodal benchmark grounded in realistic conversational dynamics. Our benchmark systematically models conversational flows derived from authentic user data, incorporating the diversity, fragmentation, and contextual dependencies observed in practice. We synthesize 1,000 high-quality dialogue turns across 100 conversations, leveraging image-question pairs from SceMQA (Liang et al., 2024) and interaction patterns from VisionArena-Chat (Chou et al., 2025). To evaluate LVLM performance, we employ a comprehensive two-way framework that combines fine-grained subdimension scoring with winrate against humanverified ground truth, achieving strong agreement with human judgments. Through extensive evaluation of state-of-the-art LVLMs, we uncover systematic weaknesses in both open-source and proprietary models, particularly in handling fragmented and context-dependent user queries. For example, even an advanced proprietary LVLM like GPT-4o sees its performance score drop from 3.92 to 2.92 on direct opening queries compared to descriptive ones. Furthermore, models struggle with iterative tasks, with GPT-4o scoring just 3.65 in refinement and 3.85 in augmentation. Smaller open-source models perform even worse in these categories, with scores as low as 1.70, highlighting a critical gap in conversational ability.

## 1 INTRODUCTION

Large Vision-Language Models (LVLMs) have rapidly evolved from simple image captioning tools into sophisticated AI assistants capable of engaging in complex, contextual conversations (Li et al., 2024; OpenAI, 2024; Bai et al., 2025; Anthropic, 2025). This progress is crucial, as users naturally gravitate towards multi-turn dialogues. This is often because non-expert users find it challenging to formulate a single, perfectly detailed instruction, and instead refine their intent and constraints iteratively over several turns (Zamfirescu-Pereira et al., 2023; Knoth et al., 2024). This conversational dynamic demands that LVLMs navigate the complexities of extended interactions, maintain context across multiple turns, and adapt to the diverse communication patterns that characterize authentic human-AI exchanges.

However, most traditional and popular benchmarks evaluate LVLMs in a fixed single-turn setting (Yue et al., 2024; Lu et al., 2024; Liu et al., 2024b), leaving an important gap between the evaluation setup and real-world application of LVLMs. To mitigate this gap, there has been a growing trend toward developing multi-turn multi-modal evaluation benchmarks (Liu et al., 2024c;a; Tian et al., 2025). However, these work largely relies on synthetic multi-turn dynamics and fail to capture the diverse dynamics in real-world multi-turn interactions. Some rely on fixed and synthetic patterns

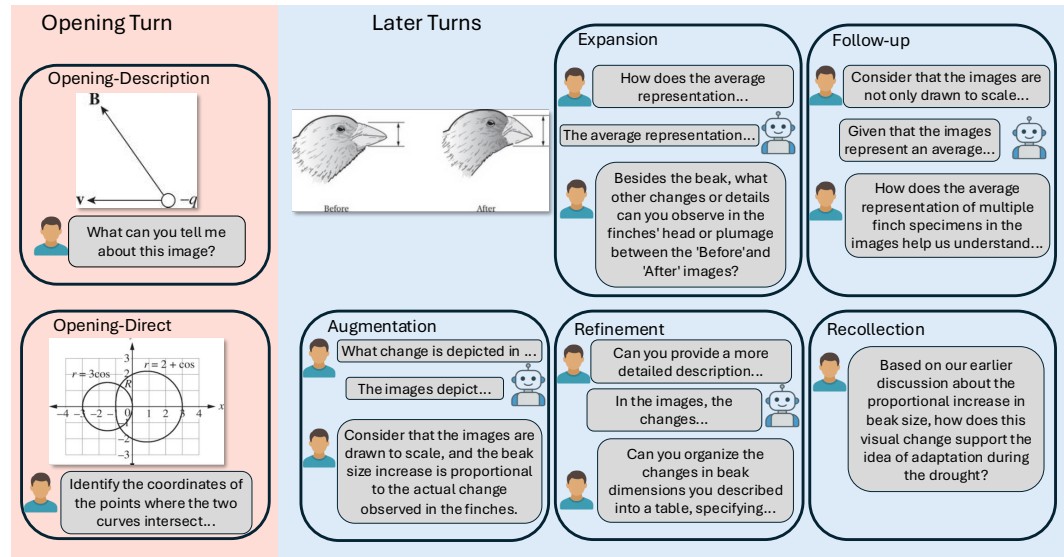

Figure 1: Taxonomy example of MMCHAT. Opening turns have two category as shown on the left. On the right is an example conversation that contains all later-turn categories of MMCHAT.

(Liu et al., 2024a), while others use less structured approaches where an LVLM autonomously generates multi-turn conversations without grounding in observed user interaction patterns (Tian et al., 2025; Liu et al., 2024c). To formally capture these conversational dynamics, a proven approach in text-only benchmarks is the development of an interaction taxonomy that summarizes diverse interaction types in a realistic distribution (Kwan et al., 2024; Li et al., 2025). However, this critical methodology remains largely underexplored in the context of LVLMs, creating a significant gap in how we evaluate their real-world performance.

In this work, we present MMCHAT: a multi-turn multi-modal conversational benchmark for LVLMs with well-defined taxonomies tailored for multi-modal interactions. Our benchmark consists of 1000 turns of conversations across 100 conversations, with image and description pairs sampled from SceMQA (Liang et al., 2024), a pre-college level STEM questions VQA dataset. We conduct an in-depth analysis of the real user-AI interaction dataset VisionArena-Chat (Chou et al., 2025), which consists of both single-turn and multi-turn conversations. We focus on the multi-turn portion of the dataset and employ human-AI dual labeling to classify the multi-turn subset of VisionArena-Chat. Our taxonomy is extended from Kwan et al. (2024) with the following modifications tailored for multi-modal conversations: For first turns, we identify two categories: **opening-direct**, where users ask specific questions or give specific instructions based on the image, and **opening-descriptive**, where users ask general descriptive questions to have the assistant describe the image first. For subsequent turns, we incorporate the following categories: **follow-up**, where the current question is directly related to a specific topic or object from previous interactions; **expansion**, where users diverge to a different topic but remain related to the image; **recollection**, where user instructions require the model to recall and retrieve information from previous turns; **refinement**, where users ask the assistant to revise or change the format of previous responses; and finally, **augmentation**, where users provide additional pieces of information or instructions to augment previous questions or instructions. We ensure our benchmark reflects the distribution of these categories by directly sampling from a transitional probability model derived from VisionArena-Chat. Using these categories and the modeled transitional probabilities, we iteratively generate each turn of the conversation with well-crafted prompts specifically designed for each category.

Our benchmark has the following distinctive features: (1) We are the first multi-turn multimodal benchmark to generate data based on both the taxonomy and conversational flow patterns of real user-AI interaction datasets. (2) Our benchmark also incorporates a largely overlooked interaction pattern where conversations begin with direct and specific queries, as opposed to descriptive ones. We observe that models tend to perform worse in these scenarios, suggesting that the ability to handle concise initial prompts is a distinct and challenging aspect of model capability that previous

evaluations may miss. (3) Our taxonomy introduces two categories, Augmentation and Refinement, to capture a critical yet overlooked dynamic in user behavior. This framework models how users iteratively build upon their instructions: they augment the context by providing new information and refine their request by adding or modifying constraints. This process of progressively revealing their full intent is particularly challenging for LVLMs, as it requires them to continuously re-ground the evolving conversation in the visual context and avoid losing sight of the image details. This align with previous findings (Laban et al., 2025)

To ensure a robust assessment, we employ a fine-grained evaluation framework. For each query category, model performance is judged across five subdimensions: two universal metrics (factual correctness and visual grounding) and three carefully designed category-specific metrics. We utilize two distinct methods for evaluation: GPT-4o provides binary scores along each subdimension, and we also calculate a win-rate against human-verified ground truth responses. The validity of our approach is confirmed by a high correlation between the overall rankings produced by both methods, further supported by a strong human-evaluator agreement rate of 0.82 for scoring and 0.76 for win-rate. These results suggest our evaluation is both comprehensive and reliable.

We evaluated ten LVLMs ranging from open-source to proprietary models. We find even the most advanced models such as GPT-4o underperform in categories like refinement and augmentation, with top models scoring as low as 3.65 in these areas. This highlights a critical inability to handle iterative, multi-turn instructions where the model must continuously re-ground the conversation in the visual context. We also detect two primary failure modes from this analysis. The first, **Conversational Decay**, describes the model's tendency to lose track of visual details and user intent over a prolonged dialogue. The second, **Error Propagation**, occurs when an initial misunderstanding, often from a direct or ambiguous opening query, cascades and magnifies through subsequent turns, leading to a complete breakdown in task fulfillment.

The contributions of our benchmark are summarized as follows:

- **A Benchmark Grounded in Real-World User Behavior:** We introduce MMCHAT, a novel benchmark with a taxonomy and conversational flows derived from authentic human-AI interactions. This captures the complexity and iterative nature of real-world use cases, which are often oversimplified in existing synthetic benchmarks.

- **A Granular, Multi-dimensional Evaluation Framework:** We propose a comprehensive evaluation framework that moves beyond simple accuracy metrics. It combines fine-grained scoring across challenging categories like **refinement** and **augmentation** with a direct win-rate comparison against human performance, enabling a deeper, more nuanced assessment of model capabilities.

- **Identification of Critical LVLM Failure Modes:** Our benchmark uncovers systematic weaknesses that previous evaluations have missed. We identify and analyze two critical failure modes—**Conversational Decay** and **Error Propagation**—and demonstrate that even state-of-the-art models struggle with the direct, specific, and iterative queries that are common in real-world interactions, challenging the validity of current evaluation approaches.

## 2 MMCHAT

### 2.1 BENCHMARK OVERVIEW

Current multi-turn multimodal benchmarks suffer from two critical limitations that compromise their ability to accurately assess LVLM capabilities in real-world settings. First, they lack well-defined taxonomies grounded in actual user-AI interactions, instead relying on either fixed synthetic patterns or unstructured LVLM-generated conversations without empirical foundation. Second, existing benchmarks predominantly use descriptive opening queries that ask models to describe images before engaging in specific tasks, which differs markedly from real user behavior and artificially simplifies subsequent reasoning tasks.

Our benchmark, MMCHAT, addresses these gaps by deriving both interaction taxonomies and conversational flow patterns directly from VisionArena-Chat (Chou et al., 2025), a large-scale dataset

of real user-LVLM interactions. We generate 1000 turns of human-verified multi-turn conversations that authentically reflect how users actually engage with LVLMs.

## 2.2 TURN INTERACTION TAXONOMY

We begin our dataset construction by conducting a comprehensive analysis of real user-AI interactions from the VisionArena-Chat dataset. To focus on substantive multi-turn conversations, we filter the dataset to include only English conversations containing 5–15 turns. This range provides sufficient context for complex interaction patterns while maintaining manageable conversation lengths for analysis.

From this filtered subset, we randomly sample a subset for detailed human annotation. Our taxonomy extends the framework proposed by Kwan et al. (2024) with specific considerations for multimodal interactions involving images.

We identify seven distinct categories of conversational turns, divided into two broad groups: first turns and later turns:

*First Turn Categories* **Opening-Direct:** Users immediately pose specific questions or provide explicit instructions directly related to visual content in the image. These turns demonstrate focused, goal-oriented interaction where users have a clear information need or task objective from the outset. **Opening-Descriptive:** Users begin conversations with broad, exploratory requests for image description or general visual analysis. These turns typically involve users seeking comprehensive understanding of the visual content before proceeding to more specific inquiries.

*Later Turn Categories* **Follow-up:** The current user input directly continues or elaborates on a specific topic, object, or concept that was the focus of immediately preceding turns. These turns maintain topical coherence and demonstrate sustained attention to particular elements within the conversation. **Expansion:** Users shift their attention to different aspects, objects, or topics within the same image while maintaining overall contextual relevance. These turns demonstrate the user's ability to explore various dimensions of the visual content systematically. **Recollection:** Users explicitly request the model to recall, reference, or retrieve specific information, details, or responses from earlier turns in the conversation history. These turns test the model's ability to maintain and access long-term conversational memory. **Refinement:** Users request modifications to the format, style, structure, or presentation of previous model responses without changing the underlying content or information. These turns focus on output optimization rather than information seeking. **Augmentation:** Users provide additional context, constraints, or instructions to enhance or modify previous responses or to re-execute previous tasks with new parameters. These turns involve iterative improvement of model outputs through supplementary user guidance. **Repetition:** Users repeat previous questions or instructions without providing any additional information, constraints, or modifications. These turns occur either due to unsatisfactory responses or to emphasize particular aspects, but crucially involve verbatim or near-verbatim restatement of earlier requests without new context or guidance. **Outliers:** Conversational turns that do not fit cleanly into the above categories, including meta-conversational comments, tangential remarks, or interactions that deviate significantly from typical multimodal conversation patterns. While infrequent, these turns capture the full spectrum of real user behavior.

## 2.3 AUTOMATED CLASSIFICATION AND TRANSITION MODELING

Following human annotation of the initial subset, we employ GPT-4o (OpenAI, 2024) to classify the remaining conversation turns in our filtered VisionArena-Chat (Chou et al., 2025) subset. The automated classification process uses few-shot prompting with human-labeled examples from each category to ensure consistency with our taxonomy definitions.

From the fully classified dataset, we compute empirical transition probabilities between turn categories (as shwon in Figure 2a), creating a Markov model that captures the natural flow of real user-AI multimodal conversations. Due to the very limited proportion of repetition and outlier categories (Figure 2b) in the dataset and their less informative nature for evaluating core conversational capabilities, we exclude these categories from the transitional modeling phase. This transitional probability matrix reveals common patterns in how users navigate between the five primary types

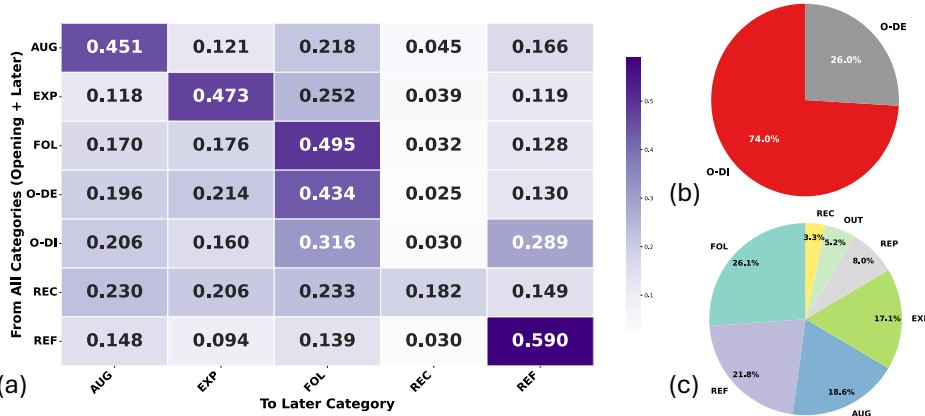

Figure 2: (a) The transition probability matrix between all turn categories (excluding repetition and outliers). (b) The distribution of opening categories for first turns. (c) The distribution of all subsequent turn categories.

of subsequent turn interactions (follow-up, expansion, recollection, refinement, and augmentation) within a single conversation, focusing on the most substantive and evaluatively meaningful interaction patterns.

## 2.4 Multi-turn Generation Pipeline

### 2.4.1 Trajectory Sampling

We begin by sampling conversation trajectories that reflect authentic user interaction patterns through a probabilistic framework. Let $C = \{c_1, c_2, \ldots, c_T\}$ represent a conversation trajectory of length $T$, where each $c_t$ denotes the category of turn $t$. We define the category sets as

$$\mathcal{S}_{\text{open}} = \{\text{Opening-Direct}, \text{Opening-Descriptive}\},$$
$$\mathcal{S}_{\text{sub}} = \{\text{Augmentation}, \text{Expansion}, \text{Follow-up}, \text{Recollection}, \text{Refinement}\},$$
$$\mathcal{S}_{\text{all}} = \mathcal{S}_{\text{open}} \cup \mathcal{S}_{\text{sub}}.$$

For the initial turn, we sample from the opening distribution:

$$c_1 \sim \text{Categorical}(\boldsymbol{\pi}_{\text{open}}), \quad c_1 \in \mathcal{S}_{\text{open}}, \tag{1}$$

where $\boldsymbol{\pi}_{\text{open}} = [0.74, 0.26]$ represents the probabilities for opening-direct and opening-descriptive categories respectively, derived from our VisionArena-Chat analysis.

For subsequent turns $t = 2, 3, \ldots, T$, we employ a Markov transition model:

$$c_t \mid c_{t-1} \sim \text{Categorical}(\mathbf{P}_{c_{t-1},:}), \quad c_t \in \mathcal{S}_{\text{sub}}, \ c_{t-1} \in \mathcal{S}_{\text{all}}, \tag{2}$$

where $\mathbf{P} \in \mathbb{R}^{7 \times 5}$ is the empirical transition probability matrix shown in Figure 2. Each row of $\mathbf{P}$ corresponds to one of the seven categories in $\mathcal{S}_{\text{all}}$, and each column corresponds to one of the five subsequent-turn categories in $\mathcal{S}_{\text{sub}}$, with rows summing to one.

The complete trajectory likelihood is thus:

$$P(C) = \pi_{\text{open}}(c_1) \cdot \prod_{t=2}^{T} \mathbf{P}_{c_{t-1},c_t}. \tag{3}$$

This process continues iteratively until we reach a predefined turn limit $T$, which we set to 10. The resulting trajectory $C$ provides a structured sequence of interaction categories that guides the content generation process while maintaining realistic conversational flow patterns derived from empirical user behavior.

### 2.4.2 Content Generation

Once a categorical trajectory is sampled, we select corresponding visual content from the SceMQA dataset (Liang et al., 2024), which provides high-quality images with associated captions and question-answer pairs covering pre-college STEM topics. Each conversation is paired with a single image-caption-QA triplet that serves as the visual foundation for all subsequent turns.

We then use GPT-4o to iteratively generate each turn's user query and assistant response along the sampled categorical trajectory using category-specific prompting strategies. Each prompt is carefully designed to maintain conversational coherence with full history context, reflect the distinct interaction patterns of its assigned category, and ensure content relevance through grounding in SceMQA visual content. Following generation, we employ human experts to verify and validate the ground truth assistant responses to ensure their correctness and quality.

### 2.5 Evaluation Pipeline

To comprehensively assess LVLM performance across our MMCHAT, we develop a category-specific evaluation framework that captures the unique requirements and challenges of each interaction type. Our evaluation employs GPT-4o as an automated judge to score model responses across multiple dimensions tailored to the conversational context.

For each turn category, we evaluate model responses using five binary evaluation dimensions that collectively provide a holistic assessment of performance. All categories share two fundamental dimensions: **factual correctness**, which verifies the accuracy of the model's response, and **visual grounding**, which ensures responses are based solely on visible image content without hallucination. Beyond these core metrics, each category incorporates three additional specialized dimensions that capture the specific capabilities required for that interaction type. Our evaluation pipeline processes each turn's question-answer pair and the dialogue history of previous turns according to its assigned category. For each evaluation dimension within a category, we craft specialized prompts that guide GPT-4o to make binary judgments (pass/fail) based on well-defined criteria. This approach yields five binary scores per turn that sum to a maximum score of 5, enabling both fine-grained analysis of specific capabilities and aggregate performance assessment across categories.

The category-specific design ensures that evaluation criteria align with the distinct expectations and challenges inherent to each interaction type. For instance, recollection turns are evaluated on their ability to retrieve and utilize information from earlier conversation history, while refinement turns are assessed on constraint compliance and incremental adaptation to new formatting requirements. Table 3 details the evaluation dimensions for each turn category, including the specific criteria used to determine pass/fail judgments for each dimension.

In addition to the scoring-based evaluation criteria, we also conduct binary winrate evaluations where judges compare model responses directly against human-verified ground truth responses. For each turn, GPT-4o determines whether the model response is preferable to or inferior to the ground truth answer. To avoid positional biases (Shi et al., 2025), we randomly swap the order of the model's response and ground truth responses during evaluation. While using GPT-4o as a judge against ground truth that it also helped generate introduces a potential for self-preference bias, this concern is mitigated by several factors. First, all ground truth responses underwent a rigorous human verification process, establishing a human-approved quality standard. Second, this win-rate method serves as a complementary signal to our dimensional scoring. The high correlation between the rankings produced by both methods confirms that any potential bias does not significantly skew the final conclusions on relative model performance.

## 3 Results

### 3.1 Experiment Setting

We evaluate the performance of both proprietary and open-sourced models on our MM-CHAT, Including GPT-4o (OpenAI, 2024), Claude-4-Sonnet (Anthropic, 2025), Qwen-2.5-VL-7B-Instruct, Qwen-2.5-VL-32B-Instruct, Qwen-2.5-VL-72B-Instruct (Bai et al., 2025), InternVL-3-

| Model | Overall | | | Category-Specific Performance | | | | | | | |
|---|---|---|---|---|---|---|---|---|---|---|---|
| | FC | VG | Winrate | Avg. | O-Di | O-De | Fol | Rec | Ref | Aug | Exp |
| *Proprietary Models* | | | | | | | | | | | |
| GPT-4o | **77.1** | **67.3** | **43.8%** | **3.95** | 2.92 | **3.92** | 4.29 | **4.35** | **3.65** | 3.85 | **4.27** |
| Claude-4-sonnet | 69.2 | 47.2 | 24.6% | 3.71 | 3.03 | 3.13 | 4.23 | 3.78 | 3.32 | 3.79 | 3.89 |
| *Open-Source Models* | | | | | | | | | | | |
| MiniCPM-o-2.6 | 40.1 | 31.8 | 13.2% | 2.46 | 1.95 | 2.08 | 2.99 | 2.32 | 1.70 | 2.66 | 2.74 |
| InternVL3-8B | 52.7 | 40.9 | 21.8% | 2.99 | 2.24 | 3.29 | 3.60 | 3.11 | 2.51 | 2.89 | 3.01 |
| InternVL3-38B | 65.4 | 51.1 | 26.4% | 3.47 | 2.42 | 3.50 | 4.19 | 3.66 | 2.97 | 3.26 | 3.79 |
| InternVL3-78B | 71.5 | 55.4 | 34.4% | 3.71 | 2.96 | 3.46 | 4.22 | 4.09 | 3.34 | 3.64 | 3.84 |
| Qwen2.5-VL-7B-Instruct | 47.5 | 33.0 | 22.5% | 2.83 | 2.39 | 2.96 | 3.52 | 3.11 | 2.13 | 2.77 | 2.85 |
| Qwen2.5-VL-32B-Instruct | 68.3 | 45.6 | 24.1% | 3.56 | **3.07** | 3.63 | 4.28 | 3.46 | 2.69 | 3.82 | 3.51 |
| Qwen2.5-VL-72B-Instruct | 71.9 | 51.2 | 35.5% | 3.73 | 2.76 | 3.54 | **4.31** | 3.84 | 2.97 | **4.07** | 3.91 |
| Mistral-3.1-24B-Instruct | 48.8 | 16.2 | 12.4% | 2.58 | 1.87 | 2.00 | 3.36 | 1.95 | 1.69 | 3.44 | 2.14 |

Table 1: **Evaluation of LVLMs on MMCHAT.** Factual correctness (FC) and visual grounding (VG) are the two common dimensions across all categories. Winrate shows performance vs human verified responses. The categories are opening-direct (O-Di), opening-descriptive (O-De), follow-up (Fol), recollection (Rec), refinement (Ref), augmentation (Aug), and expansion (Exp). Each category-specific score(0-5) is the aggregated binary score of FC and VG plus three category-specific dimensions. Avg. is the average score across all conversation turns. Highest score of each column is in blue.

8b, InternVL-3-38b, InternVL-3-78b (Chen et al., 2024), MiniCPM-o-2.6 (Yao et al., 2024). and Mistral-Small-3.1-24B-Instruct (Mistral AI, 2025).

## 3.2 MAIN RESULTS

The main evaluation results of different LVLMs are presented in Table 1. We present our main findings as follows:

**Direct Opening Queries Present Greater Challenges** Models consistently perform worse on opening-direct queries compared to opening-descriptive interactions across all evaluated LVLMs. The average performance gap ranges from 0.3 to 1.6 points, with proprietary models showing smaller but consistent differences (GPT-4o: 2.92 vs 3.92; Claude-4-Sonnet: 3.03 vs 3.13) and open-source models exhibiting more pronounced disparities (InternVL3-78B: 2.96 vs 3.46; Qwen2.5-VL-72B: 2.76 vs 3.54). This finding challenges the prevalent use of descriptive opening queries in existing benchmarks, as direct queries better reflect real user behavior patterns at the start of the conversation where users typically have specific information needs rather than seeking general descriptions. This is also supported by the real user interaction distribution in Figure 2. A qualitative example of LVLMs failing in refinement category failing in direct opening question can be found in Figure 3.

**Iterative Refinement Categories Reveal Systematic Weaknesses** Among later-turn categories, augmentation and refinement consistently emerge as the most challenging interaction types across all models. These categories, which require models to iteratively improve responses based on additional constraints or user feedback, show uniformly low performance scores. Even advanced models like GPT-4o achieve only 3.85 for augmentation and 3.65 for refinement, while smaller open-source models struggle significantly more (MiniCPM-o-2.6: 2.66 and 1.70 respectively). This pattern aligns with recent findings by Laban et al. (2025) demonstrating that LLMs "get lost" in multi-turn conversations, particularly when handling fragmented inputs and iterative refinement requests that characterize authentic user interactions. A qualitative example of LVLMs failing in refinement category can be found in Figure 3.

| Model | First-turn FC+VG Correct | | | | | | First-turn FC+VG Incorrect | |
|---|---|---|---|---|---|---|---|---|
| | All Later FC | All Later VG | Turn 2-5 FC | Turn 6-10 FC | Turn 2-5 VG | Turn 6-10 VG | All Later FC | All Later VG |
| *Proprietary Models* | | | | | | | | |
| GPT-4o | 0.87 | 0.78 | 0.87 | 0.87 (+0.00) | 0.79 | 0.78 (-0.01) | 0.63 (-0.24) | 0.49 (-0.29) |
| Claude-4-Sonnet | 0.79 | 0.56 | 0.83 | 0.77 (-0.06) | 0.66 | 0.48 (-0.18) | 0.59 (-0.20) | 0.40 (-0.16) |
| *Open-Source Models* | | | | | | | | |
| MiniCPM-o-2.6 | 0.57 | 0.49 | 0.61 | 0.53 (-0.08) | 0.53 | 0.45 (-0.08) | 0.32 (-0.25) | 0.25 (-0.24) |
| InternVL3-8B | 0.71 | 0.55 | 0.72 | 0.72 (+0.00) | 0.57 | 0.52 (-0.05) | 0.34 (-0.37) | 0.26 (-0.29) |
| InternVL3-38B | 0.76 | 0.63 | 0.73 | 0.77 (+0.04) | 0.65 | 0.61 (-0.04) | 0.56 (-0.20) | 0.40 (-0.23) |
| InternVL3-78B | 0.82 | 0.64 | 0.84 | 0.80 (-0.04) | 0.64 | 0.63 (-0.01) | 0.55 (-0.27) | 0.39 (-0.25) |
| Qwen2.5-VL-7B-Instruct | 0.60 | 0.47 | 0.66 | 0.55 (-0.11) | 0.58 | 0.38 (-0.20) | 0.35 (-0.25) | 0.21 (-0.26) |
| Qwen2.5-VL-32B-Instruct | 0.81 | 0.57 | 0.85 | 0.78 (-0.07) | 0.68 | 0.49 (-0.19) | 0.45 (-0.36) | 0.31 (-0.26) |
| Qwen2.5-VL-72B-Instruct | 0.83 | 0.64 | 0.85 | 0.81 (-0.04) | 0.70 | 0.59 (-0.11) | 0.58 (-0.25) | 0.37 (-0.27) |
| Mistral-Small-3.1-24B-Instruct | 0.52 | 0.16 | 0.57 | 0.48 (-0.09) | 0.18 | 0.14 (-0.04) | 0.40 (-0.12) | 0.09 (-0.07) |

Table 2: **Analysis of performance conditioned on first-turn accuracy and conversation length.** The table presents two scenarios. The left columns analyze conversations where the model was initially correct, measuring the performance change between early (turns 2-5) and late (turns 6-10) stages to show **conversational decay**. The right columns analyze conversations where the model was initially incorrect, with the delta quantifying the severe performance drop compared to a correct start, thereby measuring **error propagation**. Red and green indicate performance decline and improvement, respectively.

**Model Scale Correlates with Multi-turn Performance** Within the same model families, larger variants consistently outperform smaller counterparts across all conversational categories. This scaling effect is particularly pronounced in the Qwen2.5-VL series (7B: 2.83 → 32B: 3.56 → 72B: 3.73 average scores) and InternVL3 family (8B: 2.99 → 38B: 3.47 → 78B: 3.71). The improvement is most notable in categories requiring contextual reasoning and memory maintenance, such as recollection (Qwen2.5-VL: 3.11 → 3.46 → 3.84) and follow-up interactions (InternVL3: 3.60 → 4.19 → 4.22), suggesting that larger models better handle the complex contextual dependencies inherent in multi-turn conversations.

## 3.3 FURTHER ANALYSIS

Our fine-grained analysis, presented in Table 2, reveals two critical failure modes in multi-turn conversational settings: a gradual decline in performance over time, which we term **Conversational Decay**, and the catastrophic effect of initial mistakes, which we refer to as **Error Propagation**.

**Conversational Decay** Conversational Decay refers to the degradation of a model's performance in later turns of a conversation, even when its initial response is correct. As shown in the left-hand columns of Table 2, this is a prevalent issue across most models. We observe a consistent trend where both Factual Correctness (FC) and Visual Grounding (VG) scores decline when comparing early turns (2-5) to later turns (6-10).

Notably, Visual Grounding appears particularly susceptible to this decay. For instance, Qwen2.5-VL-32B-Instruct shows a substantial drop of -0.19 in VG, and even the high-performing Claude-4-Sonnet experiences a -0.18 decline. This suggests that as the conversational context becomes more complex with additional constraints and information, models struggle to consistently and accurately re-ground their responses to the visual details of the image. While Factual Correctness is generally more stable, models like Qwen2.5-VL-7B-Instruct still exhibit significant decay (-0.11). This gradual erosion of performance highlights the challenge LVLMs face in maintaining long-term context and consistent visual focus throughout extended dialogues.

**Error propagation** A more severe failure mode is Error Propagation, where an incorrect response in the first turn drastically impairs the model's ability to answer subsequent questions correctly. The two rightmost columns in Table 2 quantify this phenomenon, showing a severe performance penalty across all models.

When the first turn is incorrect, the drop in Factual Correctness and Visual Grounding is far more significant than the gradual decay observed in correct-start conversations. For example, GPT-4o's FC and VG scores drop by 0.24 and 0.29, respectively. The effect is even more pronounced for some open-source models, with InternVL3-8B showing a staggering 0.37 drop in FC and Qwen2.5-VL-32B-Instruct declining by 0.36. This indicates that once a model "hallucinates" or misinterprets the image, it is largely unable to self-correct or recover, even when subsequent turns provide an opportunity to reconsider. The initial error appears to anchor the model to an incorrect understanding, causing a cascade of failures throughout the remainder of the conversation. This highlights a critical lack of robustness and error recovery, a significant weakness for conversational agents intended for reliable, real-world deployment.

## 4 RELATED WORK

**Multi-turn Multimodal Benchmarks**  Recent efforts have introduced multi-turn evaluation for LVLMs, though with limited grounding in real user interactions. MMDU (Liu et al., 2024c) provides 1,077 multi-image conversations across 18 domains using predefined templates, while ConvBench (Liu et al., 2024a) offers hierarchical evaluation with 577 conversations following fixed synthetic patterns. MMCR (Tian et al., 2025) introduces a large-scale dataset with 310K multi-image multi-turn dialogues emphasizing contextual reasoning, though conversations are generated without grounding in specific patterns or taxonomies.

**Multi-turn Evaluation Methodologies**  MT-EVAL (Kwan et al., 2024) established foundational taxonomies for multi-turn text-only LLM interactions, providing empirically-grounded categories for information seeking, task completion, and creative collaboration. However, MT-EVAL and similar approaches like StructFlowBench (Li et al., 2025) do not address multimodal reasoning challenges or visual grounding requirements specific to LVLMs.

**Empirical Grounding and Authentic Interaction Patterns**  Current multi-turn multimodal benchmarks suffer from limited empirical grounding, as synthetic conversational scenarios fail to capture the complexity, fragmentation, and contextual dependencies of authentic human-AI exchanges (Zamfirescu-Pereira et al., 2023; Knoth et al., 2024; Laban et al., 2025). The VisionArena-Chat dataset (Chou et al., 2025) provides 230,000 real user-LVLM conversations, offering unprecedented insight into authentic multimodal interaction patterns and enabling development of more ecologically valid evaluation approaches. MMCHAT addresses these gaps by deriving both taxonomies and conversational flow patterns directly from VisionArena-Chat analysis, ensuring evaluation reflects authentic challenges faced by LVLMs in practical deployment scenarios.

## 5 CONCLUSION

In this work, we present MMCHAT, a multi-turn multimodal benchmark that addresses critical gaps in current LVLM evaluation by grounding both conversational taxonomies and interaction flow patterns in empirical analysis of real user-AI conversations from VisionArena-Chat (Chou et al., 2025). Through evaluation of 10 state-of-the-art LVLMs, we reveal systematic challenges overlooked by current evaluations, particularly models' difficulties with direct opening queries, iterative refinement processes, and fragmented user inputs that mirror real-world usage patterns.

Our findings demonstrate that even advanced models face significant challenges in multi-turn scenarios, with performance gaps particularly pronounced in augmentation and refinement categories requiring contextual reasoning. These insights align with broader research on multi-turn conversational challenges and highlight substantial work needed for robust real-world LVLM deployment. We believe MMCHAT will serve as a valuable resource for developing more capable and reliable multimodal AI systems that better reflect authentic human-AI interaction complexities.

ETHICS STATEMENT

We utilized GPT-4o (OpenAI, 2024) as an implementation tool for the construction and evaluation of our benchmark, as well as for proofreading this manuscript. Detailed usage descriptions are provided in Sections 2.3, 2.4, and 2.5, with quality control methods outlined in Appendix A. All conceptual work, including the research direction, hypotheses, and core ideas, was conducted by the authors without assistance from LLMs.

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

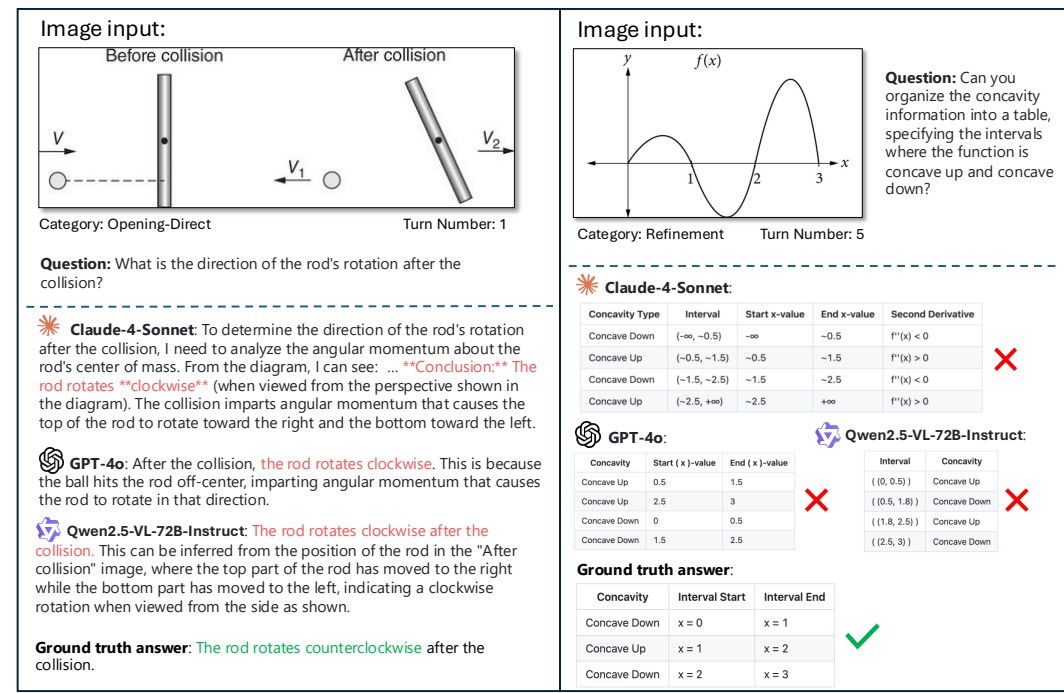

Figure 3: **Qualitative example showing LVLMs failures on MMCHAT**. The left example shows how direct opening make powerful LVLMs fail a simple visual understanding question. The example on the right shows model failures at the refinement task. The raw markdown output of the models and the ground truth are rendered for visualization purpose.

## A  QUALITY CONTROL DETAILS

### A.1  QUALITY CONTROL FOR TAXONOMY CLASSIFICATION

We employ human expert to validate all generated qa pairs by GPT-4o to make sure the ground truth reference answer is valid and correct. For the evaluation results, we sample 50 turns of conversation for each model totaling 500 turns of conversations for both scoring based and winrate evaluations. The GPT-4o as a judge results have a agreement rate of 0.82 and 0.76 respectively.

## B  EVALUATION DIMENSION DETAILS

The detailed evaluation dimensions for each category are described in Table 3. Factual Correctness and Visual Grounding are evaluated across all categories. Each category also have 3 of their own category-specific evaluation dimensions.

## C  QUALITATIVE EXAMPLE

Examples of failure cases of LVLMs can be found in Figure 3

## D  FINE-GRAINED EVALUATION RESULTS

Fine-grained evaluations of different LVLMs in category-specific dimensions are shown in Table 4 to 10.

Table 3: Category-Specific Evaluation Dimensions

| Category | Dimension | Description |
|---|---|---|
| **All Categories** | Factual Correctness | Evaluate whether the response is **factually correct** with respect to the question. |
| | Visual Grounding | Evaluate whether the response correctly **reflects visual content** without hallucinating. |
| **Opening-Direct** | Engagement Quality | Evaluate whether the opening **effectively engages** with the user's request and sets appropriate tone. |
| | Information Gathering | Assess whether the model **identifies and requests missing information** needed for a complete response. |
| | Completeness Assessment | Determine whether the model provides a **complete and thorough response** without leaving aspects unaddressed. |
| **Opening-Description** | Descriptive Depth | Evaluate whether the model provides **detailed and comprehensive descriptions** for general descriptive requests. |
| | Observational Accuracy | Assess whether the model demonstrates **careful observation skills and attention to detail**. |
| | Descriptive Organization | Evaluate whether descriptive content is structured in a **logical, coherent manner**. |
| **Augmentation** | Content Enrichment | Evaluate whether the model adds **relevant details, examples, or supporting information** that adds value. |
| | Revision Depth | Determine whether the model **thoroughly revises previous answers** rather than making surface-level tweaks. |
| | Latent Consistency | Determine whether the model maintains **logical coherence across turns** after integrating augmentation. |
| **Expansion** | Thematic Relevance | Evaluate whether the model maintains **thematic connections** when expanding to related topics. |
| | Aspect Coverage | Check whether the model addresses the **specific sub-aspect requested** rather than repeating previous information. |
| | Knowledge Integration | Evaluate how well the model **integrates new facts into established context** maintaining coherent progression. |
| **Recollection** | Selective Relevance | Evaluate whether the model **selects relevant previous elements** to recall based on the current query. |
| | Memory Accuracy | Assess whether the model **accurately recalls specific details** from earlier conversation turns. |
| | Cross-Turn Consistency | Evaluate whether the model maintains **consistency with established statements and positions**. |
| **Refinement** | Constraint Compliance | Determine if the model satisfies **all newly introduced constraints** including formatting and style requirements. |
| | Constraint Prioritization | Assess whether the model **prioritizes newer constraints over older ones** when conflicts arise. |
| | Incremental Adaptation | Evaluate whether the model **layers new constraints while preserving prior ones**. |
| **Follow-up** | Contextual Integration | Evaluate whether the model **integrates previous conversation context** in follow-up responses. |
| | Detail Elaboration | Evaluate how well the model **expands on earlier elements** with deeper insight or clarification. |
| | Progressive Understanding | Assess whether the model **builds understanding across turns** rather than treating each in isolation. |

Table 4: Opening-Direct Category Performance. FC: Factual Correctness, VG: Visual Grounding, EQ: Engagement Quality, IG: Information Gathering, CA: Completeness Assessment.

| Model | FC | VG | EQ | IG | CA |
|---|---|---|---|---|---|
| *Proprietary Models* | | | | | |
| GPT-4o (OpenAI, 2024) | 63.2 | 61.8 | 82.9 | 22.4 | 61.8 |
| Claude-4-sonnet Anthropic (2025) | 62.7 | 54.7 | 93.3 | 28.0 | 64.0 |
| *Open-Source Models* | | | | | |
| MiniCPM-o-2.6 (Yao et al., 2024) | 39.5 | 30.3 | 69.7 | 11.8 | 43.4 |
| InternVL3-8B (Zhu et al., 2025) | 51.3 | 43.4 | 65.8 | 10.5 | 52.6 |
| InternVL3-38B (Zhu et al., 2025) | 52.6 | 50.0 | 77.6 | 11.8 | 50.0 |
| InternVL3-78B (Zhu et al., 2025) | 68.4 | 61.8 | 84.2 | 18.4 | 63.2 |
| Qwen2.5-VL-7B-Instruct (Bai et al., 2025) | 47.4 | 40.8 | 78.9 | 18.4 | 53.9 |
| Qwen2.5-VL-32B-Instruct (Bai et al., 2025) | 67.1 | 52.6 | 94.7 | 23.7 | 68.4 |
| Qwen2.5-VL-72B-Instruct (Bai et al., 2025) | 61.8 | 50.0 | 86.8 | 14.5 | 63.2 |
| Mistral-Small-3.1-24B-Instruct (Mistral AI, 2025) | 23.7 | 25.0 | 65.8 | 55.3 | 17.1 |

Table 5: Opening-Description Category Performance. FC: Factual Correctness, VG: Visual Grounding, DD: Descriptive Depth, OA: Observational Accuracy, DO: Descriptive Organization.

| Model | FC | VG | DD | OA | DO |
|---|---|---|---|---|---|
| *Proprietary Models* | | | | | |
| GPT-4o (OpenAI, 2024) | 66.7 | 58.3 | 87.5 | 79.2 | 100.0 |
| Claude-4-sonnet Anthropic (2025) | 45.8 | 33.3 | 79.2 | 58.3 | 95.8 |
| *Open-Source Models* | | | | | |
| MiniCPM-o-2.6 (Yao et al., 2024) | 29.2 | 29.2 | 41.7 | 37.5 | 70.8 |
| InternVL3-8B (Zhu et al., 2025) | 58.3 | 50.0 | 75.0 | 58.3 | 87.5 |
| InternVL3-38B (Zhu et al., 2025) | 58.3 | 54.2 | 79.2 | 58.3 | 100.0 |
| InternVL3-78B (Zhu et al., 2025) | 54.2 | 50.0 | 75.0 | 66.7 | 100.0 |
| Qwen2.5-VL-7B-Instruct (Bai et al., 2025) | 54.2 | 37.5 | 66.7 | 45.8 | 91.7 |
| Qwen2.5-VL-32B-Instruct (Bai et al., 2025) | 62.5 | 45.8 | 87.5 | 66.7 | 100.0 |
| Qwen2.5-VL-72B-Instruct (Bai et al., 2025) | 66.7 | 41.7 | 79.2 | 66.7 | 100.0 |
| Mistral-Small-3.1-24B-Instruct (Mistral AI, 2025) | 100.0 | 100.0 | 0.0 | 0.0 | 0.0 |

Table 6: Follow-up Category Performance. FC: Factual Correctness, VG: Visual Grounding, CI: Contextual Integration, DE: Detail Elaboration, PU: Progressive Understanding.

| Model | FC | VG | CI | DE | PU |
|---|---|---|---|---|---|
| *Proprietary Models* | | | | | |
| GPT-4o (OpenAI, 2024) | 84.7 | 74.2 | 90.2 | 84.4 | 95.6 |
| Claude-4-sonnet Anthropic (2025) | 58.5 | 42.5 | 76.7 | 83.3 | 90.5 |
| *Open-Source Models* | | | | | |
| MiniCPM-o-2.6 (Yao et al., 2024) | 48.4 | 40.7 | 68.4 | 62.5 | 78.9 |
| InternVL3-8B (Zhu et al., 2025) | 63.7 | 51.9 | 80.9 | 76.0 | 87.4 |
| InternVL3-38B (Zhu et al., 2025) | 79.6 | 61.1 | 90.4 | 91.5 | 95.9 |
| InternVL3-78B (Zhu et al., 2025) | 79.0 | 64.4 | 92.1 | 91.0 | 95.9 |
| Qwen2.5-VL-7B-Instruct (Bai et al., 2025) | 58.5 | 42.5 | 76.7 | 83.2 | 90.5 |
| Qwen2.5-VL-32B-Instruct (Bai et al., 2025) | 81.1 | 60.0 | 93.1 | 96.7 | 97.5 |
| Qwen2.5-VL-72B-Instruct (Bai et al., 2025) | 83.6 | 63.3 | 91.6 | 95.3 | 97.5 |
| Mistral-Small-3.1-24B-Instruct (Mistral AI, 2025) | 65.5 | 19.6 | 76.0 | 88.4 | 86.9 |

Table 7: Recollection Category Performance. FC: Factual Correctness, VG: Visual Grounding, SR: Selective Relevance, MA: Memory Accuracy, CTC: Cross-Turn Consistency.

| Model | FC | VG | SR | MA | CTC |
|---|---|---|---|---|---|
| *Proprietary Models* | | | | | |
| GPT-4o (OpenAI, 2024) | 75.7 | 81.1 | 91.9 | 94.6 | 91.9 |
| Claude-4-sonnet Anthropic (2025) | 67.6 | 59.5 | 89.2 | 75.7 | 86.5 |
| *Open-Source Models* | | | | | |
| MiniCPM-o-2.6 (Yao et al., 2024) | 40.5 | 37.8 | 56.8 | 48.6 | 48.6 |
| InternVL3-8B (Zhu et al., 2025) | 51.4 | 51.4 | 68.6 | 71.4 | 68.6 |
| InternVL3-38B (Zhu et al., 2025) | 54.3 | 54.3 | 85.7 | 85.7 | 85.7 |
| InternVL3-78B (Zhu et al., 2025) | 67.6 | 73.5 | 88.2 | 91.2 | 88.2 |
| Qwen2.5-VL-7B-Instruct (Bai et al., 2025) | 54.1 | 43.2 | 73.0 | 70.3 | 70.3 |
| Qwen2.5-VL-32B-Instruct (Bai et al., 2025) | 59.5 | 54.1 | 75.7 | 78.4 | 78.4 |
| Qwen2.5-VL-72B-Instruct (Bai et al., 2025) | 67.6 | 59.5 | 89.2 | 78.4 | 89.2 |
| Mistral-Small-3.1-24B-Instruct (Mistral AI, 2025) | 35.1 | 16.2 | 40.5 | 37.8 | 64.9 |

Table 8: Refinement Category Performance. FC: Factual Correctness, VG: Visual Grounding, CC: Constraint Compliance, CP: Constraint Prioritization, IA: Incremental Adaptation.

| Model | FC | VG | CC | CP | IA |
|---|---|---|---|---|---|
| *Proprietary Models* | | | | | |
| GPT-4o (OpenAI, 2024) | 65.8 | 56.7 | 80.1 | 80.1 | 81.8 |
| Claude-4-sonnet Anthropic (2025) | 57.0 | 36.4 | 76.8 | 76.8 | 85.1 |
| *Open-Source Models* | | | | | |
| MiniCPM-o-2.6 (Yao et al., 2024) | 21.6 | 18.6 | 40.7 | 45.0 | 43.7 |
| InternVL3-8B (Zhu et al., 2025) | 39.4 | 30.6 | 59.7 | 57.9 | 63.9 |
| InternVL3-38B (Zhu et al., 2025) | 51.2 | 41.0 | 68.7 | 67.3 | 69.1 |
| InternVL3-78B (Zhu et al., 2025) | 60.3 | 43.8 | 75.3 | 73.1 | 81.3 |
| Qwen2.5-VL-7B-Instruct (Bai et al., 2025) | 33.8 | 23.4 | 48.9 | 49.8 | 56.7 |
| Qwen2.5-VL-32B-Instruct (Bai et al., 2025) | 53.7 | 32.9 | 57.1 | 56.3 | 69.3 |
| Qwen2.5-VL-72B-Instruct (Bai et al., 2025) | 55.8 | 37.7 | 65.8 | 64.5 | 73.2 |
| Mistral-Small-3.1-24B-Instruct (Mistral AI, 2025) | 31.2 | 4.8 | 39.4 | 36.8 | 57.1 |

Table 9: Augmentation Category Performance. FC: Factual Correctness, VG: Visual Grounding, CE: Content Enrichment, RD: Revision Depth, LC: Latent Consistency.

| Model | FC | VG | CE | RD | LC |
|---|---|---|---|---|---|
| *Proprietary Models* | | | | | |
| GPT-4o (OpenAI, 2024) | 82.4 | 67.0 | 75.8 | 72.5 | 86.8 |
| Claude-4-sonnet Anthropic (2025) | 76.4 | 36.3 | 93.4 | 87.9 | 85.2 |
| *Open-Source Models* | | | | | |
| MiniCPM-o-2.6 (Yao et al., 2024) | 53.3 | 34.6 | 65.4 | 48.4 | 64.8 |
| InternVL3-8B (Zhu et al., 2025) | 60.2 | 36.4 | 65.3 | 58.0 | 68.8 |
| InternVL3-38B (Zhu et al., 2025) | 69.0 | 44.8 | 73.0 | 63.8 | 75.9 |
| InternVL3-78B (Zhu et al., 2025) | 78.6 | 51.4 | 78.0 | 71.1 | 84.4 |
| Qwen2.5-VL-7B-Instruct (Bai et al., 2025) | 50.5 | 28.6 | 75.3 | 59.9 | 63.2 |
| Qwen2.5-VL-32B-Instruct (Bai et al., 2025) | 74.7 | 40.7 | 95.1 | 86.3 | 85.7 |
| Qwen2.5-VL-72B-Instruct (Bai et al., 2025) | 82.4 | 47.3 | 96.7 | 90.1 | 90.1 |
| Mistral-Small-3.1-24B-Instruct (Mistral AI, 2025) | 67.6 | 18.1 | 93.4 | 81.3 | 83.5 |

Table 10: Expansion Category Performance. FC: Factual Correctness, VG: Visual Grounding, TR: Thematic Relevance, AC: Aspect Coverage, KI: Knowledge Integration.

| Model | FC | VG | TR | AC | KI |
|---|---|---|---|---|---|
| *Proprietary Models* | | | | | |
| GPT-4o (OpenAI, 2024) | 82.3 | 71.4 | 94.9 | 88.6 | 90.3 |
| Claude-4-sonnet Anthropic (2025) | 70.7 | 49.4 | 93.1 | 87.9 | 87.9 |
| *Open-Source Models* | | | | | |
| MiniCPM-o-2.6 (Yao et al., 2024) | 39.4 | 32.0 | 75.4 | 64.6 | 62.3 |
| InternVL3-8B (Zhu et al., 2025) | 45.0 | 37.3 | 78.7 | 70.4 | 69.2 |
| InternVL3-38B (Zhu et al., 2025) | 66.5 | 54.1 | 90.0 | 83.5 | 84.7 |
| InternVL3-78B (Zhu et al., 2025) | 71.3 | 54.3 | 90.9 | 84.1 | 82.9 |
| Qwen2.5-VL-7B-Instruct (Bai et al., 2025) | 42.9 | 29.1 | 77.7 | 65.7 | 69.1 |
| Qwen2.5-VL-32B-Instruct (Bai et al., 2025) | 64.0 | 40.0 | 86.3 | 78.3 | 82.3 |
| Qwen2.5-VL-72B-Instruct (Bai et al., 2025) | 69.7 | 54.3 | 93.7 | 86.3 | 87.4 |
| Mistral-Small-3.1-24B-Instruct (Mistral AI, 2025) | 33.1 | 8.6 | 80.0 | 43.4 | 49.1 |

