# OpenReview forum: "MMChat: A Multi-turn Multi-Modal Conversational Benchmark Grounded in Real World User Behaviors"
_ICLR.cc/2026/Conference — Submitted to ICLR 2026_

### Official Review · Reviewer_PEw6 · 2025-10-27

**Soundness:** 1
**Presentation:** 2
**Contribution:** 1
**Rating:** 2
**Confidence:** 5

**Summary:**

This paper introduces MMCHAT, a multi-turn multimodal benchmark designed to address critical gaps in evaluating Large Vision-Language Models (LVLMs) by leveraging real-world user interactions. MMCHAT systematically models dialogue flows derived from authentic user data in VisionArena-Chat, incorporating diverse, fragmented, and context-dependent characteristics of human-AI conversations. The benchmark consists of 100 dialogues totaling 1000 high-quality turns, built upon image-question pairs from SceMQA and interaction patterns from VisionArena-Chat. The paper uncovers weakness in both open-source and proprietary models. The study highlights model weaknesses in handling direct opening queries, iterative refinement, and augmentation tasks, providing insights into challenges like conversational decay and error propagation.

**Strengths:**

Originality: This paper is the first to explore the dimensions of recollection, expansion, refinement, follow-up, augmentation, and repetition in the field of multi-turn multimodal evaluation.

Significance: In the field of multi-turn multimodal large language model evaluation, assessments along the dimensions of recollection, expansion, refinement, follow-up, augmentation, and repetition are highly meaningful. For example, they play an important role in advancing embodied intelligence, long-context agents, and personalized interactive systems for multimodal large models.

Clarity: The paper is easy to follow, and the stated motivation and contributions are easy to understand and relatively clear.

**Weaknesses:**

This paper is not very novel or is an incremental innovation. Reasons are as follows:

1.The identified LVLM failure mode insights are not novel and lack multimodal characteristics. Regarding the inspirations and insights obtained from the evaluation, MMChat arrived at the same conclusions as the multi-turn benchmark (MT-Eval) for large language models, namely Conversational Decay and Error Propagation. However, since this is indeed a multi-turn benchmark for multimodal large language models, evaluations conducted within these established dimensions should produce new insights to guide training.

2.Why then was only the same conclusion as MT-Eval obtained? The reason is that the organizational dimensions of the benchmark are not comprehensive. The paper directly continues using MT-Eval’s testing dimensions “recollection,” “expansion,” “refinement,” and “follow-up,” and only adds “augmentation,” “repetition,” “open-direct,” and “open-descriptive.” These additions are clearly weak and incomplete, still focusing on the detection of language capabilities within these dimensions rather than the detection of visual capabilities. When performing similar work for multimodal large language models, it is more important to emphasize the evaluation of visual information while retaining the original foundations. For example, in the augmentation pattern, beyond supplementary text, could multimodal information supplementation (e.g., sending another image) also be possible? In the recollection pattern, could it apply not just to recalling a single image but also two or even multiple images? I believe each dimension should consider multimodal aspects, aiming for maximum completeness. The current version is far from sufficient.

3.Real-world user behavior is claimed by the authors as a contribution, meaning that other multi-turn multimodal large model evaluation benchmarks are based on artificially synthesized dialogues. However, to my knowledge, ConvBench involved substantial human labor in annotation and dataset creation. Anyway, even if this is considered a contribution, the paper does not demonstrate what benefit this “real-world user behavior” brings to the evaluation. In other words, does real-world user behavior lead to different conclusions compared to artificially synthesized dialogues? The comparison with synthetic dialogues is missing. The performance comparison with previous multi-turn multimodal large model benchmarks (ConvBench, MMCR) is also missing.

4.The evaluation framework experiments are not rigorous enough. Visual grounding is entirely delegated to detection by a closed-source model, without any validation of whether the closed-source model’s visual grounding evaluation is correct, and without any comparison to human evaluation for consistency.

**Questions:**

In addition to the issues mentioned in the weaknesses, the following details are also missing.

1. Lack of key statistics of MT-Eval, including average/max turns per dialogue, average/max words in prompt, average/max words in response, average/max words per turn.

2. Lack of cost overview in using GPT4 in evaluation framework.

3. Computational resources, including the infer time for each LVLMs, evaluated time, resource consumption.

4. Effectiveness proof for evaluation framework.

**Details Of Ethics Concerns:**

Nothing

---

> ### Author Response · Authors · 2025-12-03
>
> We thank the reviewer for the constructive feedback. We address each weakness below.
>
> ### **On Weakness #1 & #2**
>
> Our taxonomy and findings are **observational**, not synthetically designed. A core contribution of MMChat is to **analyze how LVLMs fail under the actual interaction patterns that real users employ**, rather than constructing new categories top-down. Because our taxonomy is derived directly from VisionArena-Chat, some overlap with MT-Eval is expected and natural, as human conversational strategies (e.g., follow-up, refinement) are not fundamentally different between text-only and multimodal interactions.
>
> However, MMChat does extend these dimensions by incorporating **visual-grounding–dependent** patterns such as opening-direct and opening-descriptive, and by placing significant emphasis on **visual grounding** in both the construction of the benchmark and the evaluation framework. Unlike text-only settings in MT-Eval, our categories and scoring explicitly require correct grounding in the image at every turn, reflecting the multimodal nature of real LVLM use.
>
>
> Regarding the reviewer’s suggestions such as multi-image supplementation, multi-image recollection, or mid-turn image augmentation:
> **we fully agree these are important multimodal behaviors.**  However, they do not appear in the currently available real-world multimodal interaction logs. To our knowledge, **there is no publicly available large-scale dataset that contains natural multi-image conversations**. Including such behaviors synthetically would deviate from our core objective: to **faithfully model interaction patterns that users actually produce**.
>
> Importantly, our methodology is general. The taxonomy extraction, transition modeling, and turn-generation pipeline do *not* rely on single-image assumptions. Once real-world datasets with multi-image or richer multimodal interactions become available, MMChat can be extended to cover these additional behaviors with minimal modification.
>
>
> ### **On Weakness #3**
>
> Although prior benchmarks such as ConvBench involve substantial human annotation effort, their conversational structure is still **synthetically defined**: every dialogue follows a fixed three-stage pipeline (perception → reasoning → creativity), and all conversations begin with a broad descriptive query. This design does not reflect how users actually initiate or evolve multimodal interactions.
>
> In contrast, our taxonomy and turn-transition patterns are **directly extracted from VisionArena-Chat**, allowing us to model real conversational behavior. For instance, 74% of real first turns are direct, visually grounded questions rather than descriptive ones. This difference is consequential: LVLMs perform substantially worse on direct openings (e.g., GPT-4o: 2.92 vs. 3.92), a failure mode that would not be revealed in benchmarks where every conversation starts descriptively.
>
>
> ### **On Weakness #4**
>
> Our evaluation does not rely solely on a closed-source judge. All ground-truth responses are human-verified for factual correctness and visual grounding, ensuring that model outputs are always compared against a reliable, human-approved reference. We also validated judge reliability by manually inspecting a subset of GPT-4o decisions and found that its visual grounding assessments aligned with human judgment, reflected in agreement rates of 0.82 for scoring and 0.76 for win-rate.

---

> ### Author Response · Authors · 2025-12-03
>
> ### **Summary of Additional Requested Details**
>
> MMChat Dialogue Statistics
>
> | Statistic                                    | Value (Placeholder) |
> |----------------------------------------------|----------------------|
> | **Turns per dialogue**               | 10                  |
> | **Average tokens per prompt**                 | 25.02                |
> | **Maximum tokens per prompt**                 | 78                 |
> | **Average tokens per reference response**               | 103.97                  |
> | **Maximum tokens per reference response**               | 316                  |
> | **Average tokens per turn (prompt + reference response)** | 128.99               |
> | **Maximum tokens per turn (prompt + reference response)**                   | 359                  |
>
>
> All MMChat response generation were conducted on two A100 80GB GPUs using vLLM for efficient inference. The full evaluation of each model, including GPT-4o-based scoring, costs under \$50 per model. Thanks using batched api calling, each model evaluation completes in under 30 minutes.

---

### Official Review · Reviewer_z3jU · 2025-10-28

**Soundness:** 3
**Presentation:** 2
**Contribution:** 2
**Rating:** 4
**Confidence:** 3

**Summary:**

The paper introduces MMCHAT, a multi-turn, image-grounded conversational benchmark for LVLMs. The benchmark is (i) taxonomy-driven, derived from human-labeled real user–AI interactions, and (ii) flow-driven, sampling multi-turn trajectories from an empirically estimated transition model. The authors synthesize 100 conversations (1,000 turns) by pairing category-specific prompts with STEM images from SceMQA, then run a two-way evaluation: fine-grained, category-specific subdimension scores plus a win-rate vs. human-verified ground truth, both judged by GPT-4o with human verification checks and order randomization to reduce position bias. Through extensive evaluation of state-of-theart LVLMs, they have some main findings.

**Strengths:**

1.	Real-world grounding and principled taxonomy
The benchmark’s turn taxonomy (opening-direct/descriptive; follow-up, expansion, recollection, refinement, augmentation) is derived from human-annotated real dialogs and used to sample transitions that reflect authentic multi-turn distributions, rather than synthetic, fixed templates. This closes a known gap between single-turn evaluation and practical usage.
2.	Flow-aware data generation with category-specific prompting
Conversations are trajectory-sampled from an empirically estimated transition matrix and generated turn-by-turn with category-specific prompts over a shared image context, yielding coherent, long-range interactions.
3.	Clear empirical insights that matter in practice
The paper surfaces opening-direct difficulty, iterative-turn weaknesses, and detect two failure mode, decay/propagation phenomena, and per-family scaling trends—useful for model builders and eval designers.

**Weaknesses:**

1.	Judge and generator circularity / family dependence
GPT-4o is used for taxonomy auto-labeling, turn generation, and as the scoring/win-rate judge (albeit with human verification and order randomization). This creates potential circularity and family bias, especially when evaluating proprietary/open-source systems against GPT-4o-aligned criteria. The paper notes mitigation but does not quantify cross-judge robustness and reliablility (e.g., experment on different judge models to verify or estimate variance).
2.	Limited scale of curated conversations vs. source corpus
Only 100 conversations / 1000 turns are released, derived from a 230k-conversation source for flow estimation. The paper would be more enough if it show more power analysis, confidence intervals, and stability under re-sampling to show that rankings and category trends are statistically robust at this 100 size.
3.	Single-image conversations and domain coverage
Each conversation is tied to one image. While this controls variables, it narrows ecological diversity (multi-image, open-world scenes, documents/UI, egocentric video). Can the pipeline be generalized or extended to a second domain or multi-image/video scenarios?
4.	Ground-truth provenance and human verification scope
Ground truths are generated with GPT-4o then “human-verified.” The manuscript summarizes agreement rates, but it seems do not detail annotator scale, qualifications, double-blind checks, or inter-annotator agreement for ground-truth creation itself. This leaves residual uncertainty about truthfulness.

**Questions:**

See weaknesses.

---

> ### Author Response · Authors · 2025-12-03
>
> We thank the reviewer for the constructive feedback. We address each weakness below.
>
> ### **On Weakness #1**
>
> We agree that judge–generator circularity is an important concern, and we have taken care to avoid it in the design of our evaluation pipeline. While GPT-4o is involved in several steps, the benchmark is **never evaluated against GPT-4o’s own unverified outputs**. All ground-truth responses are first generated and then independently checked, corrected, or rewritten by a human annotator, which breaks any direct dependency cycle between generator and judge.
>
> In addition, we randomized the presentation order of model predictions and reference answers during win-rate evaluation to mitigate positional bias. These steps, together with human verification of judge decisions on sampled turns, help reduce potential family bias.
>
> ### **On Weakness #2**
>
> We understand the concern regarding the scale of the released benchmark. Expanding the dataset is primarily limited by the cost of human verification and evaluation, which makes it impractical to produce multiple resampled variants of the benchmark. Nevertheless, the current design provides strong statistical grounding: the benchmark includes **1,000 independently evaluated turns**, each assessed along multiple binary dimensions, resulting in a large quantity of evaluation signals per model.
>
> ### **On Weakness #3**
>
> Our use of single-image conversations follows directly from grounding the benchmark in **real user–AI interaction data**. To the best of our knowledge, **there is no publicly available largescale dataset that contains real, multi-image or video-based multi-turn conversations**. Introducing synthetic multi-image scenarios would run counter to our goal of modeling **authentic, empirically observed** conversational dynamics rather than speculative ones.
>
> At the same time, the methodology itself is general: the taxonomy derivation, transition modeling, and turn-generation framework do not rely on assumptions specific to single-image settings. Should future real user–AI datasets include richer multimodal inputs (e.g., multi-image or video), our pipeline can be extended with minimal modification.
>
> ### **On Weakness #4**
>
> The ground-truth answers were initially drafted by GPT-4o but were **strictly human-verified** before acceptance. Verification was performed by an author with a computer-science background and sufficient familiarity with the pre-college STEM content in SceMQA. Because the factual difficulty of the questions is modest, disagreements were resolved through careful re-inspection of the image and question rather than subjective interpretation.

---

### Official Review · Reviewer_pjut · 2025-10-29

**Soundness:** 3
**Presentation:** 3
**Contribution:** 3
**Rating:** 6
**Confidence:** 4

**Summary:**

This paper introduces MMCHAT, a new multi-turn multimodal conversational benchmark designed to evaluate large vision-language models (LVLMs) under realistic user interaction scenarios. MMCHAT establishes a detailed taxonomy of conversational turn types—such as opening-direct, follow-up, refinement, and augmentation. The benchmark consists of 1,000 human-verified dialogue turns (100 conversations) built upon SceMQA image–question pairs. Evaluation uses both fine-grained multi-dimensional scoring and win-rate comparisons with human responses, showing strong agreement between automated and human judgments.

**Strengths:**

**Empirically grounded design:** MMCHAT is notably based on real VisionArena-Chat data, lending ecological validity absent in prior synthetic benchmarks.

**Comprehensive taxonomy:** The paper defines clear turn categories and subdimensions, enabling systematic analysis of conversational behaviors.

**Clear paper structure and readability**: The paper is well organized and logically presented, making the methodology and results easy to follow.

**Weaknesses:**

**Limited dataset scale**: The benchmark includes only 1,000 turns, which may restrict statistical robustness and coverage across domains.

**Dependence on GPT-4o**: Using GPT-4o for both data generation and evaluation introduces potential self-bias despite mitigation efforts.

**Domain narrowness**: The use of SceMQA (STEM-oriented) content constrains conversational diversity and limits generalization to other domains.

**Questions:**

**1. Dataset Scale and Diversity:**
The benchmark currently contains 1,000 turns across 100 conversations, which seems relatively small in scale and may limit statistical robustness.

**2. Dependence on GPT-4o:**
Since GPT-4o is involved both in data generation and in the evaluation process, how did you ensure that no bias occurred? Have you validated your results using a human-only or model-agnostic judging setup?

**3.Comparison with Prior Multi-turn Benchmarks:**
Both ConvBench and MMDU are multi-turn multimodal benchmarks that also evaluate dialogue understanding over multiple images. Could you clarify in more detail how MMCHAT fundamentally differs from these prior works to substantiate the novelty and contribution of this work?

---

> ### Author Response · Authors · 2025-12-03
>
> We thank the reviewer for the constructive feedback. We address each weakness below.
>
> ### **On Weakness #1**
>
> We understand the concern regarding dataset size. The main constraint is the cost of human verification, which makes it impractical to scale to many thousands of turns while maintaining the same level of quality control. Even so, MMChat provides strong statistical grounding: it contains 1,000 independently evaluated turns, each assessed along multiple binary dimensions, resulting in a substantial number of evaluation signals per model. Importantly, the benchmark’s structure is supported by a much larger foundation, as both the taxonomy and the transition model are derived from the 230k real user–AI multimodal conversations in VisionArena-Chat. This ensures that the conversational patterns represented in MMChat reflect real usage distributions even though the curated subset is comparatively smaller.
>
> ### **On Weakness #2**
>
> Although GPT-4o participates in several steps of the pipeline, we designed MMChat specifically to avoid judge–generator circularity. The benchmark is never evaluated against GPT-4o’s unverified outputs: all ground-truth answers are independently checked, corrected, or rewritten by a human annotator, which breaks any direct dependency cycle between generation and evaluation. We also mitigate positional effects by randomizing the order in which model outputs and reference answers are presented during win-rate evaluation. Manual inspection of a sampled subset of GPT-4o’s judgments further confirmed the validity of the judge. These measures substantially reduce the likelihood of self-preference bias.
>
> ### **On Weakness #3**
>
> Our choice of SceMQA is directly motivated by the empirical distribution of the underlying VisionArena-Chat corpus from which MMChat is derived. As shown in the original data, the most common categories in real user–AI multimodal conversations are OCR, Homework, and Diagram questions—all of which closely align with the pre-college STEM content represented in SceMQA. These categories collectively account for a substantial portion of VisionArena-Chat interactions, far more than open-ended or purely creative conversation types. Grounding MMChat in this distribution therefore ensures that the curated benchmark reflects the dominant real-world usage patterns rather than introducing artificial domain diversity.
>
>
> ### **Re: Question 1**
>
> As noted above, although the curated benchmark contains 1,000 turns, the evaluation is multi-dimensional and produces a large quantity of signals per model. More importantly, the benchmark’s structure and conversational flow are grounded in a 230k-conversation real-world corpus, ensuring that the distributional properties of the interactions faithfully reflect actual user behavior.
>
> ### **Re: Question 2**
>
> We mitigate bias through a combination of human-verified ground truth, randomized answer ordering, and manual audits of judge decisions. All reference answers undergo independent human verification, preventing any direct reliance on GPT-4o’s unedited outputs. Randomizing presentation order reduces positional and stylistic preference, and manual checks confirmed that judgments followed explicit category-specific criteria.
>
> ### **Re: Question 3**
>
> MMChat differs from prior multi-turn multimodal benchmarks in several fundamental ways. It is grounded directly in real VisionArena-Chat user interactions, allowing us to derive an empirically validated taxonomy and a transition model that capture authentic conversational dynamics. This contrasts with the template-based or LVLM-generated synthetic dialogues used in ConvBench and MMDU, which do not reflect real usage distributions. MMChat also models multi-turn flow through an empirically estimated transition matrix, producing realistic conversational trajectories rather than fixed scripts. Additionally, phenomena such as opening-direct difficulty, iterative refinement and augmentation, and the failure modes of conversational decay and error propagation emerge naturally in our data-driven framework but are not represented in template-based or fully LVLM-generated datasets.

---

### Official Review · Reviewer_oPwE · 2025-10-30

**Soundness:** 2
**Presentation:** 2
**Contribution:** 2
**Rating:** 2
**Confidence:** 4

**Summary:**

MMChat introduces a multi-turn multimodal conversational benchmark grounded in real user–AI interaction data to address the gap between how large vision-language models (LVLMs) are evaluated and how they are actually used. Unlike prior benchmarks that rely on synthetic or single-turn dialogues, MMChat builds a turn interaction taxonomy from the VisionArena-Chat dataset, modeling authentic conversational patterns such as opening-direct, refinement, and augmentation turns. It synthesizes 1,000 dialogue turns across 100 conversations, paired with STEM images from SceMQA, and evaluates models using a fine-grained, category-specific framework combining factual correctness, visual grounding, and tailored subdimensions.

Extensive evaluation of 10 LVLMs reveals systematic weaknesses in handling direct queries, iterative refinement, and long-context grounding, identifying two key failure modes: Conversational Decay and Error Propagation. MMChat offers a more realistic and challenging evaluation paradigm for developing robust, context-aware multimodal AI systems.

**Strengths:**

1. The work provides a comprehensive taxonomy of seven interaction types (two opening and five later-turn categories), effectively modeling diverse user behaviors such as refinement, recollection, and augmentation, which are often ignored in previous studies.

2. The paper offers a realistic multi-turn multimodal dataset, constructed from real user–AI interaction distributions rather than purely synthetic scripts. This provides a valuable resource for evaluating LVLMs in settings closer to actual deployment.

**Weaknesses:**

1. The paper sometimes overstates its contribution. While the benchmark is grounded in real conversational statistics, the authors do not sufficiently position it relative to existing multi-turn multimodal datasets such as MMDU [1] and similar efforts. A more explicit comparison—both conceptually and empirically—would help clarify what is truly novel versus what is incremental, and would strengthen the narrative that this benchmark fills an unmet need in the space.

2. The presentation quality could be improved. Several figures and tables appear visually plain (such as Fig. 1, 2) and could benefit from clearer layout, consistent formatting, and more polished design. In addition, some equations lack alignment and visual structure (such as Line. 247, 248, 249), which affects readability. Combined with occasional clarity issues in the writing, these presentation limitations make the paper more difficult to follow than necessary and may hinder broader accessibility of the work.

3. The contribution scope feels relatively narrow given the ambition of the problem. The benchmark construction is thoughtful, but beyond dataset creation and analysis, there is limited methodological innovation.

[1] MMDU: A Multi-Turn Multi-Image Dialog Understanding Benchmark and Instruction-Tuning Dataset for LVLMs. NeurIPS 2024 Dataset & Benchmark Track

**Questions:**

1. Could you elaborate on plans to expand the dataset to include more images, domains, and longer conversations, or provide evidence that performance on this subset generalizes to other contexts?

2. Could you report inter-judge agreement between GPT-4o and human evaluators across categories?

---

> ### Author Response · Authors · 2025-12-03
>
> We thank the reviewer for the constructive feedback. We address each weakness below.
>
> ## **On Weakness #1**
>
> We agree that a clearer comparison would strengthen the work. Existing multi-turn multimodal benchmarks such as MMDU and ConvBench primarily rely on **template-driven** or **synthetically generated** conversations, where models or templates produce the entire dialogue flow without grounding in empirical user behavior. In contrast, our contribution lies in **deriving both the taxonomy and the transition structure directly from real user–AI interactions** in VisionArena-Chat. This gives us fine-grained control over what types of interactions appear and how they transition, rather than imposing fixed syntactic templates or unstructured autonomous generations.
>
> ## **On Weakness #2**
>
> We appreciate the reviewer’s observation and will improve the clarity and visual polish of the figures, tables, and equations. In the camera-ready version, we will revise Figures 1–2 for visual consistency, improve alignment and spacing in the mathematical expressions, and address minor clarity issues in the writing to ensure smoother readability.
>
> ## **On Weakness #3**
>
> We will more clearly articulate the methodological contributions. Beyond dataset construction, MMChat is the **first multimodal benchmark grounded in real conversational distributions**, and introduces:
>
> - A **real-world derived, multimodal-specific taxonomy** (not adapted from text-only settings nor synthetically defined).
> - A **probabilistic transition model** estimating conversational flow from real user behavior.
> - **Category-specific multimodal evaluation dimensions** capturing capabilities overlooked in prior work.
>
>
> ## **Re: Question 1**
>
> We designed MMChat to model **authentic** user–AI interactions. At present, **there is no publicly available dataset containing real multi-turn conversations with multi-image inputs**. Adding synthetic multi-image scenarios would deviate from our central motivation of faithfully modeling real user behaviors. However, the pipeline is general: if future real-world datasets include multi-image or video conversations, the taxonomy derivation and transition modeling can be extended with minimal modification.
>
> ## **Re: Question 2**
>
> We report the per category human evaluations results below:
>
> | Category | GPT-4o vs Human Agreement (Scoring) | GPT-4o vs Human Agreement (Win-Rate) |
> |---------|-------------------------------------|---------------------------------------|
> | Opening-Direct | 0.82 | 0.72 |
> | Opening-Descriptive | 0.84| 0.80 |
> | Follow-up | 0.72 | 0.70 |
> | Recollection | 0.92 | 0.74 |
> | Refinement | 0.90 | 0.78 |
> | Augmentation | 0.78 | 0.68 |
> | Expansion | 0.76 | 0.90 |

---

### Meta-Review · Area_Chair_pNku · 2026-01-07

**Summary:**

**Strengths**:

1. Realistic modeling of multi-turn interactions: Proposes a structured taxonomy of multi-turn multimodal interaction types (e.g., recollection, refinement, augmentation) grounded in real user–AI interaction statistics rather than purely synthetic scripts, improving ecological validity (oPwE, pjut, z3jU).

2. Flow-aware benchmark construction: Builds MMCHAT using empirically estimated transition patterns and category-aware prompting, enabling coherent multi-turn conversations that better reflect practical usage (oPwE, z3jU).

3. Practically relevant empirical insights: Identifies opening-turn difficulty, conversational decay, and error propagation phenomena that are useful for diagnosing LVLM behavior (z3jU).

**Weaknesses**:

1. Overstated novelty and contribution: Many interaction categories are inherited from prior multi-turn benchmarks (e.g., MT-Eval), with limited extension toward genuinely multimodal (especially visual) reasoning, making the contribution largely incremental (PEw6, oPwE).

2. Insufficient comparison with prior benchmarks: Lacks clear conceptual and empirical comparisons with existing multi-turn multimodal datasets such as MMDU, ConvBench, and MMCR, weakening the justification for the benchmark’s uniqueness (oPwE, pjut, PEw6).

3. Limited dataset scale and coverage: The benchmark includes only 100 conversations (1,000 turns), focuses on single-image dialogues, and is largely confined to STEM-style content, limiting statistical robustness and generalizability (pjut, z3jU).

4. Heavy reliance on GPT-4o: GPT-4o is used for data generation, auto-labeling, and evaluation, introducing potential circularity and model-family bias; cross-judge robustness and variance are not sufficiently quantified (pjut, z3jU).

5. Weak multimodal emphasis: Interaction categories mainly capture linguistic behaviors, with insufficient coverage of multimodal-specific phenomena such as multi-image reasoning, visual augmentation, or complex visual recollection (PEw6, z3jU).

6. Evaluation rigor concerns: Visual grounding is delegated to a closed-source model without validation against human judgments; key statistics, cost analysis, and computational resource reporting are missing (PEw6).

**Reviewer Concerns:**

**Addressed**:

- Insufficient comparison with prior benchmarks: Lacks clear conceptual and empirical comparisons with existing multi-turn multimodal datasets such as MMDU, ConvBench, and MMCR, weakening the justification for the benchmark’s uniqueness (oPwE, pjut, PEw6).

- Evaluation rigor concerns: Visual grounding is delegated to a closed-source model without validation against human judgments; key statistics, cost analysis, and computational resource reporting are missing (PEw6).

- Heavy reliance on GPT-4o: GPT-4o is used for data generation, auto-labeling, and evaluation, introducing potential circularity and model-family bias; cross-judge robustness and variance are not sufficiently quantified (pjut, z3jU).

**Not fully resolved**:

- Overstated novelty and contribution: Many interaction categories are inherited from prior multi-turn benchmarks (e.g., MT-Eval), with limited extension toward genuinely multimodal (especially visual) reasoning, making the contribution largely incremental (PEw6, oPwE).

- Limited dataset scale and coverage: The benchmark includes only 100 conversations (1,000 turns), focuses on single-image dialogues, and is largely confined to STEM-style content, limiting statistical robustness and generalizability (pjut, z3jU).

- Weak multimodal emphasis: Interaction categories mainly capture linguistic behaviors, with insufficient coverage of multimodal-specific phenomena such as multi-image reasoning, visual augmentation, or complex visual recollection (PEw6, z3jU).

**Reviewer Scores:**

- Reviewer oPwE: 2 -> 2
- Reviewer pjut: 6 -> 6
- Reviewer z3jU: 4 -> 4
- Reviewer PEw6: 2 -> 2

---

### Decision · Program_Chairs · 2026-01-26

Reject